# Exchange rate sensitivity influencing the economy: The case of Sri Lanka

Presant Thevakumar[1], Ruwan Jayathilaka[2]*

1 SLIIT Business School, Sri Lanka Institute of Information Technology, Malabe, Sri Lanka, 2 Department of Information Management, SLIIT Business School, Sri Lanka Institute of Information Technology, Malabe, Sri Lanka

* ruwan.j@sliit.lk

**Data Availability Statement:** This data set is publicly available to download from Central Bank of Sri Lanka (https://www.cbsl.gov.lk) or using direct access to https://www.cbsl.lk/eresearch/. The authors did not receive any special access

## Abstract

This particular study investigated the possibility of modelling the exchange rate volatility of the USD/LKR currency pair and analysed whether macroeconomic factors influence the exchange rate. To model the exchange rate volatility, a combination of Autoregressive integrated moving average (ARIMA) and generalized autoregressive conditional heteroskedasticity (GARCH) family models were used. The ARDL model was utilized to explore the presence of dynamic short-run and long-run relationships between the exchange rate and macroeconomic variables. The ARDL model empirical findings inferred that a long-run relationship does not exist between any of the examined macroeconomic variables and the exchange rate. In contrast, a short-run relationship exists between exchange rate lag one, exchange rate lag two, inflation, and merchandising trade balance. Thereby, as per the findings improving the merchandising trade balance and minimising inflation would minimise volatility in the exchange rate. All stakeholders who are exposed to foreign exchange volatility including policymakers, importers, exporters, and financial institutions can benefit from this study's findings. This research focused on the most recent economic phenomena of Sri Lanka and used Gross official reserve as a variable that was rarely used in existing literature on Sri Lankan exchange rate.

## Introduction

In a free-market economy, the exchange rate plays a crucial role in a country's volume of international trade (Import, Export). Extreme volatility in the exchange rate would directly or indirectly have an adverse impact and hinder a nation's economic progress [1]. For example, the impact of remittances on the exchange rate for developing countries is significant particularly for Nepal, India and Tajikistan [2–4]. On the contrary, for developed countries, remittance related empirical literature is almost nonexistent for contemporary Germany and the UK. Many similar studies have been conducted worldwide with different parameters to identify the factors which have a significant effect on the exchange rate. Replicability of these studies is a challenge especially when factors tested in developed countries are examined in the context of developing countries such as Sri Lanka. Only very few studies have taken place concerning the factors that impact the exchange rate in Sri Lanka. The problem with the factors associated

privileges to the data. Researchers will be able to replicate the results of this study by following the protocol outlined in the Data and methodology section of the paper.

**Funding:** The authors received no specific funding for this work.

**Competing interests:** The authors have declared that no competing interests exist.

with exchange rate volatility is that its effects are not universal and common in all countries across the globe. Several factors that affect the exchange rate in one country are not necessarily the same factors that determine the exchange rate fluctuation in another country. For instance, it is well known that the Maldives and Sri Lanka generate significant foreign exchange earnings from tourism. The COVID-19 pandemic globally impacted including these two countries. However, the Maldives tourism bounced back within a year. In contrast, Sri Lanka is still struggling to revive tourism in 2022.

Firms in Sri Lanka that have an international presence by either importing or exporting, are exposed to the volatility risk of the exchange rate. In 1977, Sri Lanka relaxed its economy for international trade, which is known as the 'open economy'. As a result, it created a favourable climate for foreign direct investments, which was induced by providing investment incentives, export-led growth and receiving financial assistance from international financial institutions [5]. A fixed exchange rate regime was practised until 1978 November 15 after that until 2001, the Central Bank of Sri Lanka maintained a floating rate regime. From November 2001 onwards, the island nation has been practising a floating rate regime.

Investigating the sensitivity of exchange rate volatility is useful in identifying the direction of exchange rate movements and consequently, their repercussions on the economy.

An understanding of the exchange rate movements accurately, and reliably as well as modelling are imperative for a nation's policymakers to formulate the appropriate monetary policy and fiscal policy for a given country. This is because unpredictable excessive volatility would deter investors to invest in the country due to monetary instability [6, 7]. It will also cause a continuous rise in the prices of imports. Hence, accurate modelling of the volatility would help stakeholders at all levels to anticipate the exchange rate risk and prepare proactively with risk management approaches.

## Objectives

The research aims to consider the impact of macroeconomic factors in influencing the exchange rate in Sri Lanka. Thus, this study varies and is unique in two dimensions. Firstly, to model exchange rate volatility using ARIMA and GARCH models. Secondly, to identify the major factors that impact the exchange rate of Sri Lanka, especially to analyse the relationship between the Sri Lankan Rupee (LKR) and the United States Dollar (USD). Although a few studies have been conducted in Sri Lanka, these studies have either examined multiple variables for a short period or have examined one variable for a long period [8–11]. Moreover. This study will focus on the newly available up-to-date data and information and accordingly will analyse the factors which significantly influence volatility. This publication will carry unique findings that will assist the policymakers to deal proactively with monetary and fiscal policies as well as firms that carry out imports or export through the international market.

The remainder of the paper is organised as follows. Section 2 provides a brief overview of the relevant literature, conceptual frameworks, and hypothesis development. The methodology is presented in Section 3. Section 4 assesses the empirical findings and Section 5 presents the conclusion and policy implications.

## Literature review

Existing literature related to the Exchange rate in Sri Lanka was analysed to identify relevant variables and priority of variables. The purpose of the analysis was to build our research on existing Sri Lankan exchange rate literature. Firstly, not only macro-economic variables, but also other factors such as the nation's investment climate, political stability, and media outlook could influence the exchange rate. Thus, many consider modelling exchange rate volatility

using statistical models as a better tool than solely relying on macroeconomic fundamentals. Hence a suitable exchange rate volatility model for Sri Lanka was analysed. Secondly, The impact of macro-economic variables on Sri Lanka was investigated. Based on available Sri Lankan literature: remittance, gross official reserve, merchandise trade balance, and money growth macro-economic variables were selected. In addition to the above three variables, inflation and interest rates' impact on the exchange rate impact were also analysed since the impact of these variables on the exchange rate is widely accepted.

## Modelling exchange rate volatility

Perera and Rathnayaka [11] have investigated appropriateness of the model to forecast exchange rate volatility in Sri Lanka using factors such as exports, imports, remittance, tourist arrivals and Colombo share market movements. The empirical analysis was performed based on data from January 2010 to December 2017. From multiple models, it was identified that ARIMA (1,0,0)-GARCH (1,0) is the ideal model for the analysed period. Another similar study was performed by KandeArachchi [12] who investigated exchange rate volatility using data from Jan 2000 to August 2018 using the daily exchange rate and inferred that the best model for the analysed period was AR(2)-GARCH(1,1). Since the literature indicates that volatility can be analysed using a volatility model, this research has also attempted identify a suitable volatility model for exchange rate volatility.

## Remittance and exchange rate

Rajakaruna [13] conducted a study in Sri Lanka to examine the factors that influence exchange rate volatility in Sri Lanka. Here, multiple regression models and vector autoregression models were employed. The sample data analysis period was from 2001 to 2010. According to this study, the variables net foreign purchase, official intervention, and interest rate have a major impact on the exchange rate. However, the two techniques utilised by the researcher indicated a discrepancy in results which raised the question- i.e. whether the multiple regression or VAR model is appropriate for Sri Lanka or maybe there could be another better technique that could provide better results than Multiple linear regression and VAR.

## Foreign reserves and exchange rate

A study by Ariyasinghe and Cooray [14] investigated the nexus between foreign reserves and inflation in Sri Lanka. The sample period for data collection of the study was based on monthly statistics from January 2003 to March 2020. In this study, the ARDL model, bound testing techniques, Vector autoregression, error correction, and Johansen cointegration techniques were employed. According to the empirical findings, the exchange rate indicated a cointegration relationship between inflation and foreign reserves in the long run. This finding confirmed the vitality of reserves that the higher accumulation of foreign reserves could reduce inflation for small open economies such as Sri Lanka.

## Money supply growth and exchange rate

A time-domain study was conducted by Maitra [15] on money supply and exchange rate in Sri Lanka using cointegration and VEC based on monthly data from 2001-to 2008. Here, the empirical findings suggested that the seventh and nine lags of the money supply caused depreciation in the exchange rate.

### Foreign trade and exchange rate

It is commonly accepted that the balance of trade has an impact on volatility in exchange rates. A study conducted by Senadheera [16] investigated the effect of the exchange rate on the trade balance in Sri Lanka. The study sample period included quarterly data from 2000 to 2013. The study's empirical findings confirmed that depreciation of the nominal effective exchange rate will deteriorate the trade balance in the long run. Moreover, this study sheds light on how growth in money circulation in the economy can be detrimental to a country's trade of balance in the long term. The lesson learnt from this study in the context of Sri Lanka is that devaluing the domestic currency (LKR) to attract investments would be harmful in the long run. Increasing the money supply and depreciating the currency should not be viewed as gateways for new investments.

## Data and methodology

This section has two subsections; the first subsection examines how data were collected from various reliable sources. The second subsection provides detailed information regarding the methodology and the reasoning for following the chosen methodology.

### Data

Average monthly exchange rate data were collected from the Central bank of Sri Lanka's (CBSL) website. Similarly, the monthly inflation rate (NCPI) Year-on-Year in percentage data, Monthly Policy interest rate (Average weighted call money rate) data, Monthly Remittance (in USD Million) data, Monthly Gross official reserve (in USD Billion) data, Monthly Merchandise trade balance (In USD) data and, Monthly broad money growth (%, M2B) data were collected from CBSL website. Potential variables were identified through literature review and availability of the data.

### Research framework and hypothesis

Six hypotheses are derived based on the literature review. Fig 1 provides an overview of the conceptual framework for this study.

The main hypotheses are summarised as follows:

$H1_1$. Exchange rate volatility is dependent on Inflation.

$H2_1$. Exchange rate volatility is dependent on the Interest rate.

$H3_1$. Exchange rate volatility is dependent on Remittances.

$H4_1$. Exchange rate volatility is dependent on Gross official reserve.

$H5_1$. Exchange rate volatility is dependent on Money supply growth.

$H6_1$. Exchange rate volatility is dependent on the Merchandise trade balance.

This study will conduct empirical testing and will either accept or reject the null hypothesis based on the empirical findings and their statistical significance.

### Fitting a model

Initially, before any model could be fitted the exchange rate data should be stationary. Therefore, a stationarity test would be conducted using the Augmented dicker fuller test and Philip perron test. Once we have completed the test, we can infer that the data is stationary in either level form or first difference. Thereafter, we can fit an ARIMA model or ARCH and GARCH models.

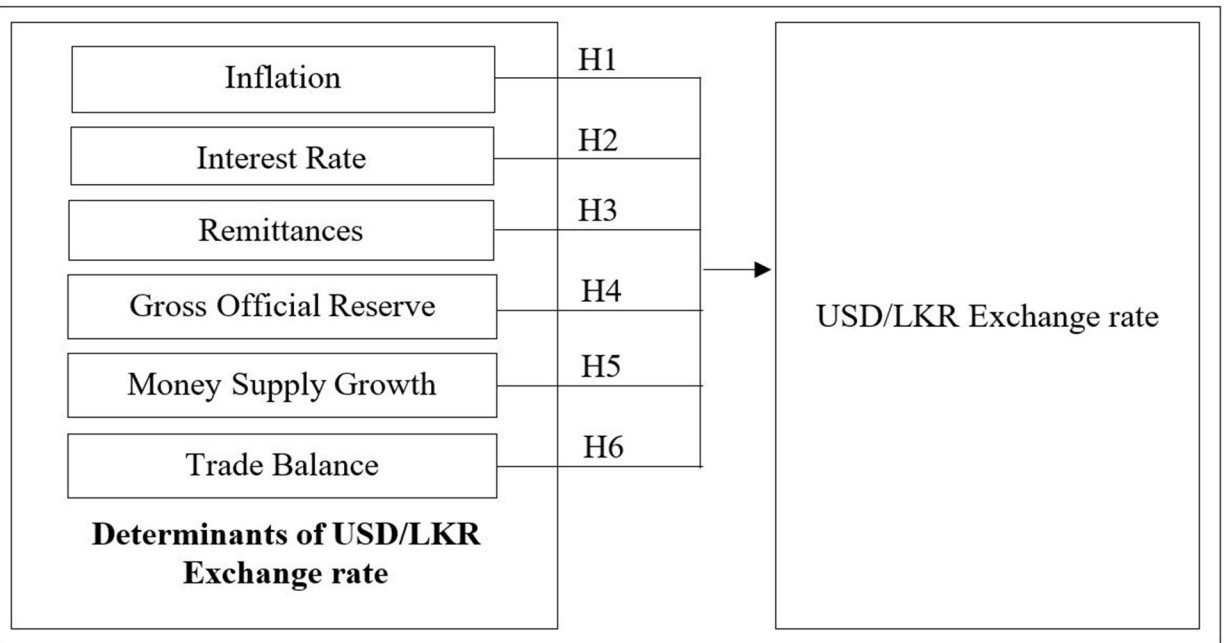

**Fig 1. Conceptual framework.** Source: Authors' construct based on the reviewed literature.

## ARIMA(p,d,q) model

ARIMA is one of the Box and Jenkins [17] methodologies. Using this approach we try to find the ARIMA (p,d,q) model which describes the stochastic process from the existing sample data. identify the following is correlated with past value autoregression or moving average first the optimal lag would be identified using correlogram and partial correlogram and next the several ARIMA models would be tested to identify the best fitted ARIMA model. The Akaike information criterion (AIC) would be used to select the optimal model the model with the lowest AIC value would be selected. The ARIMA (p,d,q) equation can be expressed as follow:

$$\varphi_p \ (L) \ (1-L)^d(y-\mu) = \vartheta_q \ (L) \ e_t \tag{1}$$

where

$\varphi_{p=1-} \sum_{i=1}^{p} \varphi_i L^i \ and \ \vartheta_q \ (L) = 1 - \sum_{i=1}^{q} \vartheta_j L^j$ are polynomials in terms of $L$ of degree p and q. $y_t$ is the time series, and $e_t$ is the random error at time $t$, with $\mu$ being the mean of the model. $d$ is the order of the difference operator. $\varphi_1 \ \varphi_2 \ \varphi_3$ and $\vartheta_1 \ \vartheta_2 \ \vartheta_3$ are the parameters of autoregressive and moving average terms with order p and q respectively.

$L$ is the difference operator defined as $\Delta y_t = y_t - y_{t-1} = (1-L) \ y_t$

The optimal lag selection for ARIMA will be estimated using the graphical correlogram and partial correlogram technique. Thereafter will proceed further with volatility model fitting.

## ARCH(p) model

The basic Autoregressive Conditional Heteroskedasticity (ARCH) has two-equation a conditional mean equation and a conditional variance equation. This model was developed by Engle [18] for examining the volatility of financial time series ARCH (p) model equation can be

expressed as follow:

$$\varepsilon_t = z_t\, \sigma_t \text{ where } z_t \text{ is where noise}$$
$$\sigma^2 t = \omega + \sum_{i=1}^{p} \alpha_i \varepsilon_{t-1}^2 \qquad (2)$$

where $\omega > 0 \geq 0,\ \alpha_i > = 0$ and $i > 0$

## GARCH(p,q) model

The Generalized Autoregressive Conditional Heteroskedasticity(GARCH) developed by Bollerslev [19] is an extended version of the ARCH model that allows the error variance to be dependent on its past lag value.

The GARCH (p,q) model can be expressed as follow:

$$\sigma^2{}_t = \omega + \sum_{i=1}^{p} \alpha_i \varepsilon_{t-1}^2 + \sum_{j=1}^{q} \beta_j \sigma_{t-j}^2 \qquad (3)$$

The assumption is that for every $p \geq 0$ and $q > 0$, the parameters are unknown and since the variance is positive, then the following relations must be positive also $\omega \geq 0$, and $\alpha_i \geq 0$ for every $i = 1,..,q$ and $\beta_j \geq 0$ for $j = 1,...,p$.

## ARCH LM test

Firstly, before fitting either an ARCH or GARCH model, the examining data should have an ARCH effect. The ARCH -LM test which was developed by Engle [20] can be used to check whether the ARCH effect exists or not. If the data exhibits the ARCH effect, we could also proceed with the Fitting GARCH family model else another model should be applied. once the ARCH effect has been verified we could apply the ARCH, GARCH family model could be fitted with optimal lag selection. To identify whether the model is acceptable, the log-likelihood should be higher, and it should withstand the diagnostic test such as the Godfrey Autocorrelation test. This test checks whether Autocorrelation exists on the residuals, which is a prerequisite to applying the GARCH family models. Furthermore, a Heteroskedasticity test would be conducted for the residuals and the prior expectation is that the residuals should be heteroskedastic. In addition, the Jarque Berra normality test and Breusch-Pagan white test would be performed. Accordingly, the best volatility model will be selected.

## Analysing the impact of macroeconomic variables

This section examines whether a dynamic causal relationship exists between exchange rate (dependent variable) and macroeconomic variables (Independent variables) in Sri Lanka using the ARDL bound testing approach to cointegrate for the period from January 2009 to May 2021. Before any test can be performed all the time series data should remain stationary. The Augmented Dickey-Fuller test and the Philip Perror test were performed to ensure all variables are stationary All the variables should either be integrated in the order I (0) or I (1) to apply the ARDL bound test. If the series is not stationarity findings would be insignificant. The ARDL was selected because with this even when exogenous variables are integrated in a different order I (0) or I (1), researchers could apply the same. In contrast, to apply VECM or ECM all variables should be integrated in the same order. The AIC will be used to select the optimal lag of variables. Thus, this study will use ARDL to examine the long-run and short-run relationship between exchange rate and macroeconomic variables.

Firstly, the Optimal lag of the variable for cointegration will be selected using the AIC criterion. Thereafter, a short-run relationship will be tested and next the bound test and long-run relationship will be tested. Following the guidelines of Pesaran, Shin [21], the ARDL equation

**Table 1. Summary of descriptive statistics.**

| Variable | Mean | SD | Min | Max | Skewness | Kurtosis |
|----------|------|-----|------|------|----------|----------|
| EXRATE | 142.46 | 25.85 | 109.5 | 197.63 | 0.55 | 2.13 |
| DEXRATE | 0.56 | 1.72 | 4.59 | 8.32 | 1.49 | 7.94 |
| INF | 5.11 | 2.45 | -0.71 | 10.44 | -0.01 | 2.28 |
| IR | 6.95 | 1.03 | 4.5 | 10.5 | -0.12 | 5.28 |
| REM | 517.24 | 116.5 | 241.34 | 812.73 | -0.58 | 2.81 |
| GOR | 6771.58 | 1607.5 | 1454.71 | 9935.77 | -1.22 | 5.24 |
| MG | 16.87 | 3.91 | 6.60 | 22.94 | -0.65 | 2.81 |
| TB | 0.66 | 0.22 | -1.10 | -0.08 | 0.47 | 2.68 |

Source: Authors' computation based on the collected data.

for this study is given as follows:

$$\Delta EXRATE_t = \beta_0 + \sum_{i=1}^{m} \beta_1 \Delta EXRATE_{t-1} + \sum_{i=1}^{m} \beta_2 \Delta INF_{t-1} + \sum_{i=1}^{m} \beta_3 \Delta IR_{t-1}$$
$$+ \sum_{i=1}^{m} \beta_4 \Delta REM_{t-1} + \sum_{i=1}^{m} \beta_5 \Delta GOR_{t-1} + \sum_{i=1}^{m} \beta_6 \Delta MG_{t-1}$$
$$+ \sum_{i=1}^{m} \beta_7 \Delta TB_{t-1} + \sum_{i=1}^{m} \alpha_1 \Delta EXRATE_{t-1} + \sum_{i=1}^{m} \alpha_2 \Delta INF_{t-1} + \qquad (4)$$
$$\sum_{i=1}^{m} \alpha_3 \Delta IR_{t-1} + \sum_{i=1}^{m} \alpha_4 \Delta REM_{t-1} + \sum_{i=1}^{m} \alpha_5 \Delta GOR_{t-1}$$
$$+ \sum_{i=1}^{m} \alpha_6 \Delta MG_{t-1} + \sum_{i=1}^{m} \alpha_7 \Delta TB_{t-1} + \mu_t$$

The letter "$m$" denotes the optimum lag selected by AIC criteria. The coefficient $\beta_{0 \text{ to }} \beta_7$ represents short-run relationship whereas the $\alpha 1$ to $\alpha 7$ coefficient represents the long-run relationship. If all the coefficient of $\alpha 1$ to $\alpha 7$ is equal to zero it means which is the null hypothesis failing to reject it implies that there is no long-run relationship.

## Results and discussions

Table 1 summarises descriptive statistics of the exchange rate and macroeconomic variables. The exchange rate (*EXRATE*), exchange first difference (*DEXRATE*), and the trade balance (*TB*) all have positive skewness. Positive skewness indicates that the tail on the right is longer than the tail on the left which means the bulk of the data values lies left of the mean value. By contrast, inflation (*INF*), interest rate (*IR*), remittance (*REM*), Gross official reserve (*GOR*), and money growth (*MG*) all have negative skewness; these imply that the tail on the left is longer than the tail on the right and the bulk of data lie right of the mean value.

Apart from the first difference in the exchange rate, Interest rate, and Gross official reserve all other variables have Kurtosis less than 3. This finding implies that these variables have lighter tails than normal tails. On the contrary, the first difference between exchange rate, Interest rate, and the gross official reserve has a fatter tail than the normal distribution.

### Volatility model estimation

This section examines the best volatility model for USD/LKR exchange rate volatility. Before conducting any test, we should ensure the collected time series data is stationary to avoid spurious regression. To test stationarity, graphical analysis, and the formal test the Augmented Dickey-Fuller test and Philip Perron test were used. The average monthly USD/LKR data was used for data analysis which is denoted by the *EXRATE*.

Fig 2(A) exhibits average monthly exchange rate graphs; it can be inferred that the *EXRATE* does not exhibit stationarity and has no mean reversion. Fig 2(B) shows the exchange rate first

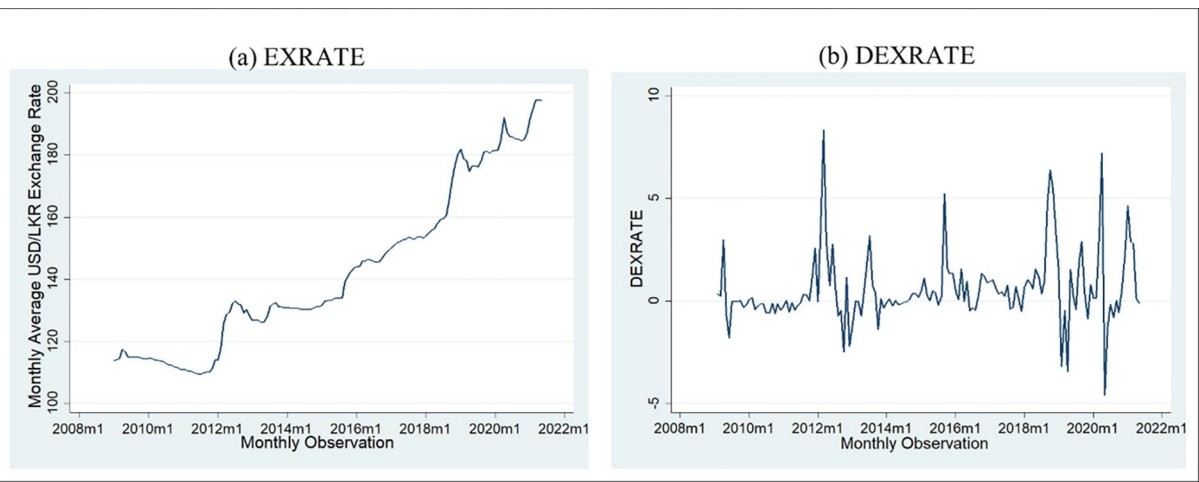

**Fig 2. Exchange rate and exchange rate difference time series graph.** Source: Authors' construct based on the collected data.

difference monthly graph, stationarity and means reversion. Hence, the exchange rate first difference can be utilised for statistical testing purposes. Table 2 tests the stationary of these two variables using the formal Augmented Dickey-Fuller test and Philip Perron Test.

According to Table 2, both ADF and PP tests confirm that *EXRATE* is not stationarity in the level form however it is stationary at the first difference with a significance of 0.01 hence for further data analysis the first difference will be utilised is denoted with *DEXRATE*.

According to Fig 3(A), Optimal lag selection to fit an ARIMA model for MA lag one, and Fig 3(B) for autoregression, lag one was identified to be outside of the confidence bound with higher-order magnitude.

According to Table 3, using the AIC criterion and SC criterion the best ARIMA model fitting for the Exchange rate is ARIMA (1,0,0).

Hence ARIMA (1,0,0) will be used to identify the autoregression of the *EXRATE*. Table 4 indicates that Autoregression is statistically significant with a Log-likelihood of 277.63. The authors will further examine whether an ARIMA and ARCH family combination model could produce a better fitting model. To examine ARCH family models, authors need to validate that the ARCH effect is present in the data series. The ARCH LM test is utilised to test the ARCH effect.

The results in Table 5 suggest that the ARCH effect is present from lag 1 to lag 8. Hence, we reject the null hypothesis as there is no ARCH effect.

Since we have identified ARCH effect is present, it is feasible to further proceed with the GARCH family model the best-fitted volatility model to model LKR/USD exchange rate volatility.

## ARIMA-ARCH family estimation

According to Table 6, The best-fitted model is ARIMA (1,0,0)-ARCH (1) with the normal distribution which has a log-likelihood of -249.13, Although the T-distribution for all 3 models produced a high log-likelihood, the parameters were insignificant. The authors also conducted using GED distribution, however, neither of these produced an outcome.

Analysis proved that all parameters are statistically significant. Some diagnostic tests were performed to determine the statistical significance of the above mentioned models. Firstly, Godfrey's autocorrelation test was performed on the residuals. The findings suggest that

**Table 2. Stationarity test of average USD/LKR exchange rate.**

| Variable | | Unit Root Tests | | | |
|---|---|---|---|---|---|
| | | ADF with Drift | | PP Test with one lag | |
| *EXRATE* | | Test stat | P-Value | Test Stat | p-value |
| | Level | 0.73 | 0.76 | 1.31 | 0.99 |
| | First Diff | -6.57*** | 0.00 | -7.94 | 0.00 |

Source: Authors' computation based on the data.

Notes

*** denotes a 0.01 level of significance.

residuals do have autocorrelation. The results are significant with an alpha of 0.01. Autocorrelation was tested for all the models. Secondly, Jarque Bera's Normality test was performed. The results suggest that the data is not normally distributed. Many researchers propose that the GARCH family model can be accepted even if they are not normally distributed [19]. Hence, we could accept this model. The Breusch-Pagan heteroskedasticity test suggests that the residuals are heteroskedastic. Despite ARIMA-EGARCH (1,1) producing a lower log-likelihood when the Godfrey autocorrelation test was performed to determine whether autocorrelation exists as a prerequisite to the GARCH family models to be significant its results suggested that autocorrelation does not exist hence this model is statistically insignificant. Although the ARIMA (1,0,0)-GARCH (1,1) produces a higher log-likelihood, the GARCH parameter is not significant hence this model is not the appropriate fit.

## Cointegration method

The Cointegration technique was also employed to determine whether Macroeconomic variables influence the exchange rate. Before conducting the cointegration, all the variables should be in either order of integration I (0) or I (1). No variable should be integrated in the order I (2) to determine the order and stationarity of the variables ADF test, PP tests were performed. According to the stationarity test result, all variables are integrated of order I (0) except for EXRATE which is integrated into I (1). Fig 4 below from (a) to (f) show the graphical

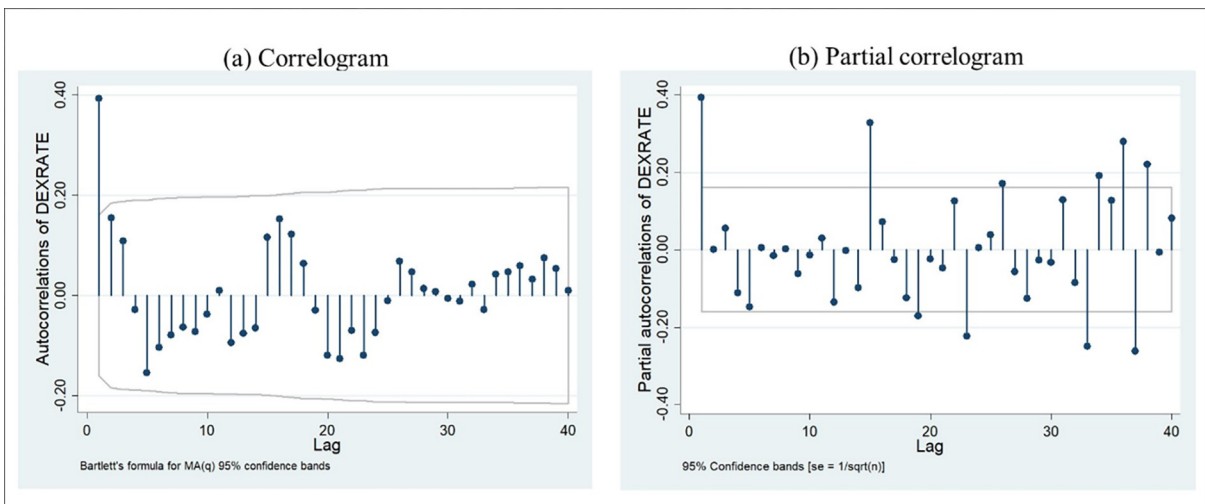

**Fig 3. Correlogram and partial correlogram of exchange rate.** Source: Authors' construct based on the collected data.

**Table 3. Feasible ARIMA models.**

| ARIMA model | AIC | SC |
|---|---|---|
| *DEXRATE* | | |
| (0,0,1) | 561.5646 | 567.55903 |
| (0,1,1) | 582.59013 | 588.571 |
| (1,0,0) | **559.26945** | **565.26387** |
| (1,0,1) | 561.26924 | 570.26088 |
| (1,1,0) | 595.26144 | 601.24231 |
| (1,1,1) | 560.46721 | 569.4385 |

Source: Authors' computation.

representation of the variables. As per the graphical representation from Fig 4(A)–4(F), all variables exhibit stationarity to confirm that this statistical stationarity test will be conducted. Fig 2(A) and 2(B) already confirms that the Exchange rate is nonstationary and that the exchange rate first difference is stationary.

Table 7 depicts those variables integrated into the level order except for EXRATE. Even when one variable is integrated in different order only the ARDL test could be performed. To conduct the test, the optimal order of the variables needs to be selected. According to our conceptual framework exchange rate is the dependent variable and all the other variables are independent exogenous variables.

The most suitable lag as per Table 8 is lag two. Thus, the ARDL test will be performed with a lag order of two. To identify the optimal lag AIC, HQIC and SBIC pointed out that lag two is the appropriate optimal lag.

The ARDL test was performed with an optimal lag of two. According to Table 9, the bound test suggests that the dependent variables and the exogenous variable do not exhibit any long-run relationship at all levels. As a result, the only short-run relationship will be examined.

According to Table 10, short-run relationships exist. The exchange rate lag one, exchange rate lag two, inflation, and the net trade balance are statistically significant. The Adjusted R-Square of 0.9964 means 99.64% of the variability in the exchange rate could be explained by the statistically significant variables. Inflation and exchange rate show a negative and statistically significant 0.01 level relationship in the short run with a coefficient of 0.15. This means that a 1% increase in the inflation rate would depreciate the LKR/USD exchange rate by 0.15

**Table 4. Estimation of ARIMA (1,0,0) model.**

| D. | OPG | | | |
|---|---|---|---|---|
| *D.EXRATE* | Coef. | Std. Err | z | P>\|z\| |
| _cons | 0.007 | 0.006 | 1.16 | 0.244 |
| ARMA | | | | |
| AR | | | | |
| L1 | 0.38*** | 0.059 | 6.40 | 0.000 |
| MA | | | | |
| L1 | -0.99 | 287.9413 | -0.00 | 0.997 |
| /sigma | 1.57 | 226.39 | 0.01 | 0.49 |

Source: Authors' computation.

Notes

*** denotes a 0.01 level of significance.

**Table 5. ARCH LM test.**

| lags(p) | chi$^2$ | df | Prob > chi$^2$ |
|---------|---------|-----|----------------|
| 1 | 18.009 | 1 | 0.0000 |
| 2 | 18.390 | 2 | 0.0001 |
| 3 | 18.401 | 3 | 0.0004 |
| 4 | 18.488 | 4 | 0.0010 |
| 5 | 18.492 | 5 | 0.0024 |
| 6 | 18.542 | 6 | 0.0050 |
| 7 | 18.730 | 7 | 0.0091 |
| 8 | 18.868 | 8 | 0.0156 |

Source: Authors' computation.

per cent, keeping other variables constant. Initially, Dornbusch [22] discussed the linkage between inflation and exchange rate and developed an econometric model to analyse the impact of exchange rate on prices. Our findings are consistent with the work of Asari, Baharuddin [23] who proved that exchange rate and inflation have an inverse relationship in Malaysia.

Considering trade balance, it exhibits a short term negative 0.001 level significance relationship with the exchange rate. An increase of 1% in net trade balance would result in the LKR depreciating against the USD by 2.15%. The most likely explanation for this finding is that Sri Lanka most of the time had a trade deficit. This means that Sri Lanka as a developing economy should focus on reducing the trade deficit and transform into an economy that exports essential products or those with an inelastic demand (where the demand is less sensitive in terms of volatility) leading to a trade surplus.

Thus, the LKR would appreciate this setting. This finding is consistent with the research of Senadheera [16] who examined the trade balance and exchange rate in Sri Lanka and inferred that depreciation of the currency would be detrimental to trade in the long run. These findings are evident in Sri Lanka and a similar scenario have been continuing for decades, with the depreciation of the LKR thus pushing to further widen the trade deficit. When considering exchange rate past values, the short-run relationship indicates that the current exchange rate has a significant 0.001 level relationship between its past lag one and lag two values. The exchange rate lag one value coefficient is 1.28; this means a 1% appreciation in the previous day's exchange rate would appreciate today's exchange rate by 1.28%. The lag two exchange rate coefficient is -0.28, which means an appreciation of 1% in the previous two days' exchange rate would depreciate the exchange rate by 0.28%. These findings are consistent with the study results of Alagidede and Ibrahim [24] who found out that exchange volatility is self-driven and only a smaller proportion could be explained by macroeconomic variables in Ghana.

## Conclusion and policy implications

The purpose of this research paper is to explore a reliable model to assess exchange rate volatility and determine whether any macroeconomic variables contribute to influence USD/LKR exchange rates. The importance of this research area is that the study findings provide insights into the exchange rate volatility which indicates the overall health of the economy. Exchange rate instability could directly impact investors' confidence in a country and as a result, could deter any new investments; To attract investments, Sri Lanka should be perceived as a destination conducive to foreign investments. The data analysis was performed for the period from January 2009 to May 2021. The best-fitted model to fit volatility was identified to be ARIMA

**Table 6. Estimated ARIMA-GARCH family models.**

| ARIMA (1,0,0)-ARCH (1) | |
|---|---|
| Parameter | Coefficient (Normal) |
| ω | 0.89*** |
| α1 | 0.83*** |
| Log-likelihood | **-249.1334** |
| Jarque-Bera (Normality) | 457.7*** |
| Breusch-Pagan (Heteroskedasticity) | 0.00*** |
| Wald chi2 | 114.94*** |
| Godfrey test (Autocorrelation) | 7.37*** |
| ARIMA (1,0,0)-GARCH (1,1) | |
| Parameter | Normal |
| ω | 0.96*** |
| α1 | 0.83*** |
| β1 | -0.35 |
| Log-likelihood | -247.65 |
| Jarque-Bera | 442.4*** |
| Breusch-Pagan | 0.00*** |
| Wald chi2 | 92.93*** |
| Godfrey test | 6.68*** |
| ARIMA (1,0,0)-EGARCH (1,1) | |
| Parameter | Normal |
| ω | 0.06 |
| α1 | 0.55*** |
| β1 | 0.84*** |
| γ1 | 0.074 |
| Log-likelihood | -240.62 |
| Jarque-Bera | 247.7*** |
| Breusch-Pagan | 0.00*** |
| Wald chi2 | 15.89*** |
| Godfrey test | 0.47 |

Source: Author's computation based on the data.

Notes

*** denotes 0.01 level of significance

**denotes 0.05 level of significance and

* denotes 0.1 level of significance.

(1,0,0)-ARCH (1) model; the alpha was 0.833, highly statistically significant, and it also satisfied all the diagnostic tests. Series exhibits autoregression and the significance of the ARCH parameter imply that the exchange rate is influenced by long-term memory and shocks affecting the exchange rate will be permanent and do not disappear. Thus, the current and future value of the exchange rate depends on the past value of the exchange rate. Hence, the policymakers must adopt a farsighted approach concerning the decision making of monetary and fiscal policies. They need to be vigilant and refrain from implementing policies that create sudden shocks in the exchange rate which would exacerbate the depreciation of the LKR against the USD.

Furthermore, the study explored whether dynamic causality relationships exist between exchange rate and macroeconomic variables. The empirical findings suggested that there is no

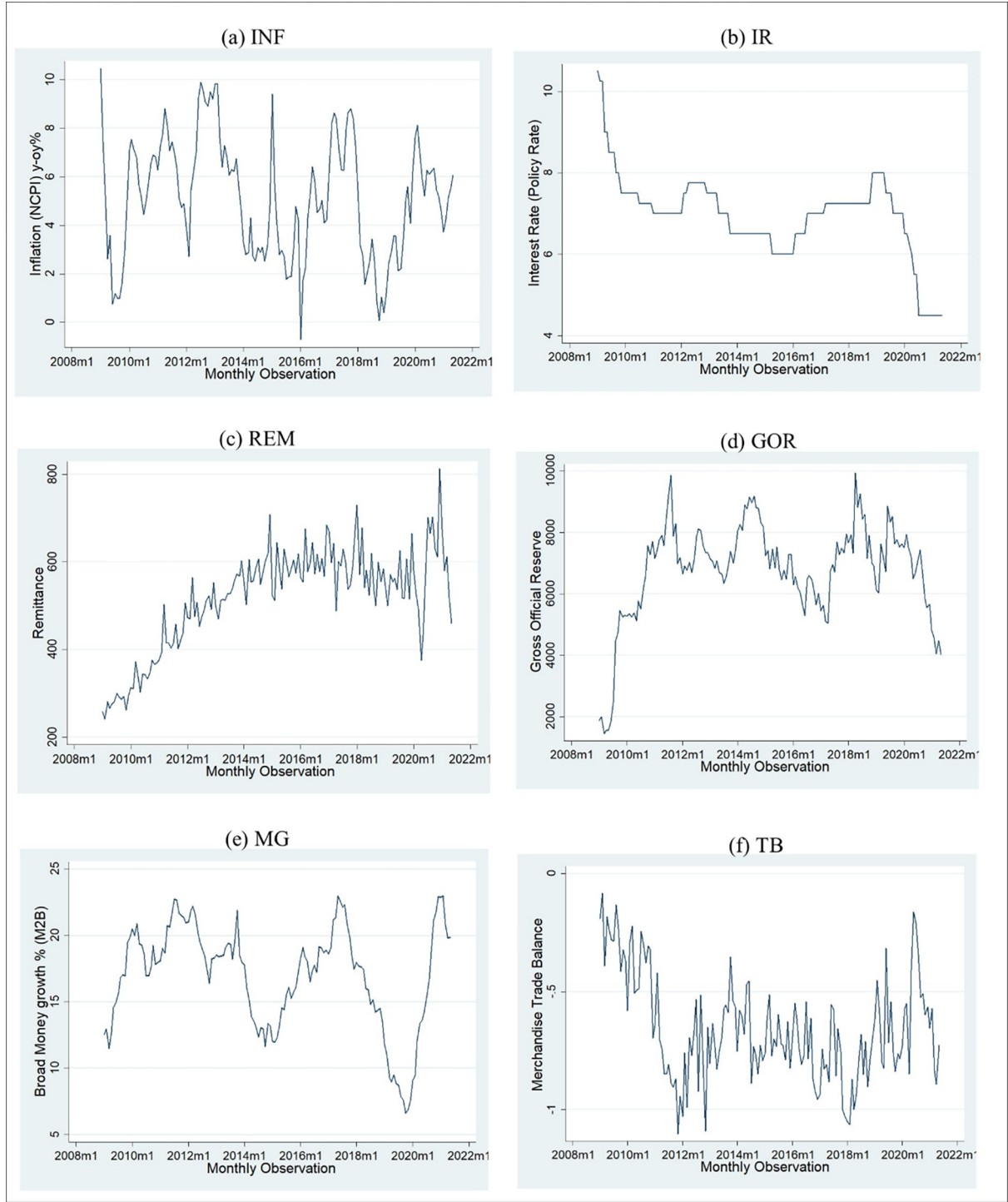

**Fig 4. Stationarity time series of the examining variables.** Source: Authors' construct based on the collected data.

long-run relationship between exchange rate and macroeconomic variables, However, test findings indicated that a short-run relationship exists between exchange rate and exchange rate lag one and lag two, inflation and net trade balance. Interestingly remittance, interest rate,

**Table 7. Stationarity test of the variables.**

| Variable | | ADF with Drift | | PP Test with one lag | | |
|---|---|---|---|---|---|---|
| | | Test stat | p-value | Test Stat | p-value | Integrated order |
| EXRATE | Level | 0.73 | 0.76 | 1.33 | 0.99 | I(1) |
| | First Diff | -6.57 | 0.00 | -7.94 | 0.00 | |
| INF | Level | -3.95 | 0.00 | -3.86 | 0.00 | I(0) |
| | First Diff | -8.29 | 0.00 | -10.3 | 0.00 | |
| IR | Level | -2.37 | 0.01 | -2.57 | 0.09 | I(0) |
| | First Diff | -6.39 | 0.00 | -12.167 | 0.00 | |
| REM | Level | -3.02 | 0.00 | -3.4 | 0.00 | I(0) |
| | First Diff | -12.62 | 0.00 | -18.57 | 0.00 | |
| GOR | Level | -2.89 | 0.00 | -3.031 | 0.03 | I(0) |
| | First Diff | -7.78 | 0.00 | -15.402 | 0.00 | |
| MG | Level | -2.04 | 0.02 | -1.856 | 0.35 | I(0) |
| | First Diff | -6.58 | 0.00 | -9.811 | 0.00 | |
| TB | Level | -4.08 | 0.00 | -5.11 | 0.00 | I(0) |
| | First Diff | -13.24 | 0.00 | -18.92 | 0.00 | |

Source: Authors' computation based on the collected data.

Notes

*** denotes 0.01 level of significance, **denotes 0.05 level of significance, and *
denotes 0.1 level of significance.

gross official reserve, and money supply growth are statistically insignificant for the period of study.

The uniqueness of this research is a suitable volatility model was identified which represents the contemporary economic situation of Sri Lanka. The researchers agree that based on changes in the political, economic, social, technological, and legal environment, the suitability of the proposed volatility model could become obsolescence and the situation could warrant a different model. However, having a model to cope with short-term, and medium-term uncertainty could assist investors to make well-informed investment decisions. Another unique component of the research is the macro-economic variables which were selected to analyse their short-term and long term impact on the exchange rate. From numerous possible factors which could influence the exchange rate, this research paper attempted to explain the impact of foreign reserves, money supply, remittances, inflation and interest rate on the exchange rate using monthly data for the last 12 years.

**Table 8. Lag order selection.**

| lag | LL | LR | df | p | FPE | AIC | HQIC | SBIC |
|---|---|---|---|---|---|---|---|---|
| 0 | -618.78 | | | | 328.241 | 8.63155 | 8.68994 | 8.77526 |
| 1 | -268.96 | 699.64 | 1 | 0.00 | 2.67098 | 3.82021 | 3.88694 | 3.98444 |
| 2 | -262.38 | 13.16* | 1 | 0.00 | 2.47326* | 3.74326* | 3.81833* | 3.92802* |
| 3 | -262.29 | 0.18 | 1 | 0.67 | 2.5046 | 3.75579 | 3.8392 | 3.96108 |
| 4 | -261.82 | 0.93 | 1 | 0.33 | 2.52326 | 3.76314 | 3.8549 | 3.98896 |

Source: Authors' computation based on the collected data.

Notes: Endogenous: EXRATE and Exogenous: INF, IR, REM, GOR, MG, TB.

**Table 9. Bound test result for cointegration.**

| Significance | Critical Value Bounds | |
|---|---|---|
| | **I0 Bound** | **I1 Bound** |
| 10% | 6.58 | 6.58 |
| 5% | 8.21 | 8.21 |
| 2.5% | 9.80 | 9.80 |
| 1% | 11.79 | 11.79 |
| F-Statistics | 0.297 | |

Source: Authors' computation based on the collected data.

The empirical findings can give rise to the following policy implications. Firstly, the exchange rate depends on its past values which means that any monetary shocks will have a lasting impact on the exchange rate. Those who are keen to invest in Sri Lanka would always prefer a stable currency regime that can sustain the real value of LKR, with a likelihood of stability in profitability, minimise financial risks etc. Hence, this finding can guide potential stakeholders to make reliable and well-informed investment decisions. Furthermore, local firms involved in international trade could make timely decisions about timelines for exports and as well as imports. Based on the current exchange movements, business firms could hedge against the risk of excessive exchange rate volatility. Secondly, inflation and trade balance has a considerable impact on the exchange rate. Therefore, policymakers could manage the

**Table 10. Short-run relationship.**

| Variable | Coefficient | St. Error | t-Statistics |
|---|---|---|---|
| Exrate(-1) | 1.28*** | 0.07 | 16.36 |
| Exrate(-2) | -0.28*** | 0.08 | -3.55 |
| INF | -0.15*** | 0.06 | -2.62 |
| IR | -0.019 | 0.18 | -0.11 |
| REM | -0.00 | 0.00 | -0.16 |
| GOR | -0.00 | 0.00 | -0.76 |
| MG | 0.047 | 0.04 | 1.07 |
| TB | -2.15*** | 0.74 | -2.87 |
| Constant | -0.80 | 2.63 | -0.31 |
| Dependent variable = Exrate | | | |
| Exogenous variables = INF, IR, REM, GOR, MG, TB | | | |
| Log likelihood = -267.5917 | | | |
| R- Square = 0.9966 | | | |
| Adjusted R-Square = 0.9964 | | | |
| **Diagnostic Test** | | | |
| Jarque bera—174.4 | | | |
| Godfrey test—0.086(0.7696) | | | |
| White's Test—96.62 (0.0000***) | | | |

Source: Author's computation based on the data.

Notes

*** denotes 0.01 level of significance

**denotes 0.05 level of significance and

* denotes 0.1 level of significance.

exchange rate by controlling inflation to an acceptable level and by sustaining the affordability of Sri Lanka's exports in the eyes of other nations as buyers. By helping to minimise price fluctuations, buyers are likely to form a better and more favourable perception of Sri Lanka's exports. Moreover, this type of scenario enables the retention and securing of export buyers. Thus, it would facilitate a trade surplus and the LKR appreciate against the USD. Interestingly, controlling the exchange rate by interest rate and remittance does not seem to have a significant impact in the case of Sri Lanka.

## Acknowledgments

The authors would like to thank Ms. Gayendri Karunarathne for proof-reading and editing this manuscript.

## Author Contributions

**Conceptualization:** Ruwan Jayathilaka.

**Data curation:** Presant Thevakumar.

**Formal analysis:** Presant Thevakumar, Ruwan Jayathilaka.

**Investigation:** Presant Thevakumar, Ruwan Jayathilaka.

**Methodology:** Presant Thevakumar, Ruwan Jayathilaka.

**Resources:** Ruwan Jayathilaka.

**Software:** Presant Thevakumar, Ruwan Jayathilaka.

**Supervision:** Ruwan Jayathilaka.

**Validation:** Presant Thevakumar, Ruwan Jayathilaka.

**Visualization:** Presant Thevakumar, Ruwan Jayathilaka.

**Writing – original draft:** Presant Thevakumar, Ruwan Jayathilaka.

**Writing – review & editing:** Ruwan Jayathilaka.

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
