## [Decision Letter · Decision Letter 0]

10 May 2022

PONE-D-21-40958Exchange Rate Sensitivity Influencing the Economy: The Case of Sri LankaPLOS ONE

Dear Dr. Jayathilaka,

Thank you for submitting your manuscript to PLOS ONE. After careful consideration, we feel that it has merit but does not fully meet PLOS ONE’s publication criteria as it currently stands. Therefore, we invite you to submit a revised version of the manuscript that addresses the points raised during the review process.

We look forward to receiving your revised manuscript.

Kind regards,

Ricky Chee Jiun Chia

Academic Editor

PLOS ONE

Journal Requirements:

Reviewers' comments:

Reviewer's Responses to Questions

**Comments to the Author**

1. Is the manuscript technically sound, and do the data support the conclusions?

Reviewer #1: Partly

2. Has the statistical analysis been performed appropriately and rigorously? 

Reviewer #1: Yes

3. Have the authors made all data underlying the findings in their manuscript fully available?

Reviewer #1: Yes

4. Is the manuscript presented in an intelligible fashion and written in standard English?

Reviewer #1: No

5. Review Comments to the Author

Reviewer #1: The article "Exchange Rate Sensitivity Influencing the Economy: The Case of Sri Lanka" (PONE-D-21-40958) has been an interesting read, but I feel that it should be improved at several, different points:

1) there are many grammatical errors and/or typos and/or repetitions and/or missing plurals: "This present reseach", "not used earlied used in Sri Lankan studies", "the ability borrow", "Foreign direct investment", "one of the most discussed microeconomic variables", "foreign reserve", "explored the determinates", "could cause deter" etc. I think that the whole paper should liguistically revised;

2) the contrast between some parts of the paper and others is particularly strong. On the one hand, the authors define macroeconomic concepts, which are rather known and unnecessary ("Exchange rate or conversion rate simply means the..." or "The exchange rate volatility means that the prices of imports and exports fluctuate" or "The reason for selecting USD to analyse the exchange rate is because the majority of international trade is conducted as USD"). On the other, they all of the sudden run several econometrical tests which are in turn particularly complex. The jump from one approach to another is - in my opinion - too pronounced and several, rather self-evident definitions should be cut and/or better articulated;

3) I don't really understand why the authors have to create a complex research scheme based on several papers (whose "suitability" as explained in Fig. 1 cannot be externally verified), although the macroeconomic factors identified as traditionally explaining exchange rate volatility are commonly accepted and do not need to be really justified. For instance, the relationship between interest rates or inflation and exchange rates has been widely analysed in the economic literature, meaning that it is not necessary to base the authors' decision on some papers retrieved in electronic journal databases. Moreover, I am pretty sure that much more articles have been written on these topics than those selected. Therefore, the parts from page 5 to 10 are - at least, in my opinion - not necessary or should be reduced so that the basic ideas (i.e., why the authors will analyse specific macroeconomic factors) are conveyed without getting too much into the details of the papers (which mention other countries than Sri Lanka (i.e., are therefore distracting));

4) I feel that the authors should add graphs (for instance, with data from World Bank databases) showing the evolution of the macroeconomic factors analysed for Sri Lanka during the time period of analysis. For instance, the exchange rate LKR/USD, the inflation rate etc.;

5) the authors should better explain the added-value of their research. What is new? What is particular? These are mentioned in the "Objective", but should be better highlighted also in the "Conclusion";

6) some assertions should be better clarified by means of economic literature. For instance, from "Although numerous studies..." to "fluctuation in another country".

I feel, in general, that the paper should be reduced in its length while making its key messages clearer.

6. PLOS authors have the option to publish the peer review history of their article (what does this mean?). If published, this will include your full peer review and any attached files.

Reviewer #1: No

---

## [Author Response · Author response to Decision Letter 0]

21 May 2022

Point–by–point response to reviewers

Dear editor and reviewer.

Greetings. Thank you very much for the fruitful comments.

Comments from Authors: Please note that page numbers and line numbers refereed in this document is align with the revised manuscript which has track changes.

Comments of Reviewer:

Comment #1: There are many grammatical errors and/or typos and/or repetitions and/or missing plurals: "This present reseach", "not used earlied used in Sri Lankan studies", "the ability borrow", "Foreign direct investment", "one of the most discussed microeconomic variables", "foreign reserve", "explored the determinates", "could cause deter" etc. I think that the whole paper should linguistically revised;

Comments of Authors:

Thank you and well noted. This has been in cooperated in the revised manuscript with track changes.

• "This present reseach" - rectified in Line #23.

• "not used earlied used in Sri Lankan studies" rectified line # 41.

• "the ability borrow” - rectified line #73.

• "Foreign direct investment" rectified line #71.

• "one of the most discussed microeconomic variables" – removed – line # 192.

• "foreign reserve" - rectified in line # 252, 253, 268, 269.

• "could cause deter" – rectified Line #619.

The specifically highlighted errors have been corrected. In addition, revised version has been proofread. Correction has been made including typos, comma correction and brevity.

Comments of Reviewer:

Comment #2: The contrast between some parts of the paper and others is particularly strong. On the one hand, the authors define macroeconomic concepts, which are rather known and unnecessary ("Exchange rate or conversion rate simply means the..." or "The exchange rate volatility means that the prices of imports and exports fluctuate" or "The reason for selecting USD to analyse the exchange rate is because the majority of international trade is conducted as USD"). On the other, they all of the sudden run several econometrical tests which are in turn particularly complex. The jump from one approach to another is - in my opinion - too pronounced and several, rather self-evident definitions should be cut and/or better articulated;

Comments of Authors:

Thank you and well noted. This has been in cooperated in the revised manuscript with track changes.

• "Exchange rate or conversion rate simply means the..." – removed Line #47-48.

• "The exchange rate volatility means that the prices of imports and exports fluctuate" – removed. Line # 78 -82.

• "The reason for selecting USD to analyse the exchange rate is because the majority of international trade is conducted as USD – Removed. Line # 98-101.

Comments of Reviewer:

Comment #3: I don't really understand why the authors have to create a complex research scheme based on several papers (whose "suitability" as explained in Fig. 1 cannot be externally verified), although the macroeconomic factors identified as traditionally explaining exchange rate volatility are commonly accepted and do not need to be really justified. For instance, the relationship between interest rates or inflation and exchange rates has been widely analysed in the economic literature, meaning that it is not necessary to base the authors' decision on some papers retrieved in electronic journal databases. Moreover, I am pretty sure that much more articles have been written on these topics than those selected. Therefore, the parts from page 5 to 10 are - at least, in my opinion - not necessary or should be reduced so that the basic ideas (i.e., why the authors will analyse specific macroeconomic factors) are conveyed without getting too much into the details of the papers (which mention other countries than Sri Lanka (i.e., are therefore distracting));

Comments of Authors:

Thank you for your informative feedback. We agree with your statement. Literature review has been revised in the manuscript with track changes from Line #113 to 305 only to include Sri Lankan related studies and not to delve in too much detail of each study.

Comments of Reviewer:

Comment #4: I feel that the authors should add graphs (for instance, with data from World Bank databases) showing the evolution of the macroeconomic factors analysed for Sri Lanka during the time period of analysis. For instance, the exchange rate LKR/USD, the inflation rate etc.;

Comments of Authors:

Thank you and well noted. We would like to incorporate World Bank data however World Bank database only provides Inflation either on an Annual basis (URL1) or quarterly basis (URL2). Since we analysed variables on a monthly basis, we were unable to incorporate World Bank data. Therefore, we were limited to only Monthly data from Central Bank of Sri Lanka and have accordingly created graphs of all the analysed variables in Page No 17.

Comments of Reviewer:

Comment #5: the authors should better explain the added value of their research. What is new? What is particular? These are mentioned in the "Objective", but should be better highlighted also in the "Conclusion";

Comments of Authors:

Thank you and well noted. This has been in cooperated in the revised manuscript with track changes from Line # 653 to 663.

Comments of Reviewer:

Comment #6: some assertions should be better clarified by means of economic literature. For instance, from "Although numerous studies..." to "fluctuation in another country".

Comments of Authors:

Thank you and well noted. With regards to “numerous countries” relevant example and explanation is given from Line # 50 to 58

With regards to "fluctuation in another country" relevant example and explanation is given from Line # 63 – 67.

---

## [Editor Report · Decision Letter 1]

24 May 2022

Exchange Rate Sensitivity Influencing the Economy: The Case of Sri Lanka

PONE-D-21-40958R1

Dear Dr. Ruwan Jayathilaka,

We’re pleased to inform you that your manuscript has been judged scientifically suitable for publication and will be formally accepted for publication once it meets all outstanding technical requirements.

Kind regards,

Ricky Chee Jiun Chia

Academic Editor

PLOS ONE
---

## [Editor Report · Acceptance letter]

1 Jun 2022

PONE-D-21-40958R1 

Exchange Rate Sensitivity Influencing the Economy: The Case of Sri Lanka 

Dear Dr. Jayathilaka:

I'm pleased to inform you that your manuscript has been deemed suitable for publication in PLOS ONE. Congratulations! Your manuscript is now with our production department. 

Kind regards, 

on behalf of

Dr. Ricky Chee Jiun Chia 

Academic Editor

PLOS ONE